# Tissue Expression and Variation of the *DGAT2* Gene and Its Effect on Carcass and Meat Quality Traits in Yak

**DOI:** 10.3390/ani9020061

**Published:** 2019-02-14

**Authors:** Jiang Hu, Bingang Shi, Jianpeng Xie, Huitong Zhou, Jiqing Wang, Xiu Liu, Shaobin Li, Zhidong Zhao, Yuzhu Luo

**Affiliations:** 1Faculty of Animal Science and Technology & Gansu Key Laboratory of Herbivorous Animal Biotechnology, Gansu Agricultural University, Lanzhou 730070, China; huj@gsau.edu.cn (J.H.); 15293103434@163.com (B.S.); 18709480641@163.com (J.X.); huitong.zhou@lincoln.ac.nz (H.Z.); wangjq@gsau.edu.cn (J.W.); liuxiu@gsau.edu.cn (X.L.); lisb@gsau.edu.cn (S.L.); zhaozd@gsau.edu.cn (Z.Z.); 2Gene-Marker Laboratory, Faculty of Agriculture and Life Sciences, Lincoln University, Lincoln 7647, New Zealand

**Keywords:** *DGAT2*, variation, carcass trait, meat quality trait, tissue expression, yak

## Abstract

**Simple Summary:**

Yaks (*Bos grunniens*) inhabit the Qinghai-Tibetan Plateau and adjacent highlands at elevations between 2000 and 5000 m, where they are important domestic animals, as they provide meat, milk, fuel, and other necessities for Tibetans and nomads in China. Yak meat is fine in texture and high in protein, yet poor in muscular marbling and tenderness. Diacylglycerol acyltransferase-2 (*DGAT2*), which regulates fat deposition in animals, is a candidate gene for meat quality and quantity traits. However, there have been few reports on the effects of the *DGAT2* gene on the meat quality of yak. Our study elucidated tissue expression of the yak *DGAT2* gene and association of variation in the gene with Warner–Bratzler shear force of longissimus muscle. The results provide guidance for the molecular-assisted selection of meat tenderness in yak.

**Abstract:**

Diacylglycerol acyltransferase-2 (*DGAT2*) plays a key role in the synthesis of animal triglycerides (TGs). This study investigated the relative expression of the *DGAT2* gene in tissues, variation in the gene, and its association with carcass and meat quality traits in yaks (*Bos grunniens*). *DGAT2* was found to be expressed in twelve tissues investigated, but the highest expression was detected in subcutaneous fat, and moderate levels were observed in the liver, heart, *longissimus dorsi* muscle, and abomasum. Three variants (*A_1_ to C_1_*) were found in intron 5 and another three variants (*A_2_ to C_2_*) were found in intron 6, with two single-nucleotide polymorphisms (SNPs) being identified in each region in 694 Gannan yaks. Variants *B_1_* and *C_2_* were associated with a decrease in Warner–Bratzler shear force (WBSF) (*p* = 0.0020 and *p* = 0.0441, respectively), and variant *C_1_* was associated with an increase in WBSF (*p* = 0.0434) and a decrease in drip loss rate (*p* = 0.0271), whereas variant *B_2_* was associated with a decrease in cooking loss rate (*p* = 0.0142). Haplotypes *A_1_-A_2_* and *B_1_-A_2_* were found to be, respectively, associated with an increase and a decrease in WBSF (*p* = 0.0191 and *p* = 0.0010, respectively). These results indicate that *DGAT2* could be a useful gene marker for improving meat tenderness in yaks.

## 1. Introduction

Triglycerides (triacylglycerols, TGs) are the major energy storage molecules in most eukaryotic cells. The enzyme Acyl-CoA: diacylglycerol acyltransferase (DGAT), which was first reported in chicken liver in 1960 [1], plays a predominant role in catalyzing the final, rate-limiting step of triglyceride synthesis in mammals [2]. Two DGAT enzymes, *DGAT1* and *DGAT2*, which are involved in different pathways, have been identified in a wide variety of eukaryotes [3,4,5].

The existence of *DGAT2* was predicted from the finding that *DGAT1*-deficient mice (*Dgat1^−/−^*) were viable and still able to synthesize TGs [6]. *DGAT2* belongs to an acyltransferase gene family that has no homology with *DGAT1* [5] and is highly expressed in tissues that make large amounts of TGs, including the liver, white adipose tissues, and mammary glands [7,8], and appears to be predominantly responsible for TG homeostasis in vivo [9]. Therefore, *DGAT2* is a promising candidate gene for traits related to lipid synthesis and storage in farm animals.

The bovine *DGAT2* maps to chromosome 15q25–q26 [10] and numerous variations have been detected in cattle [10,11,12]. The expression of *DGAT2* is positively correlated to intramuscular fat (IMF) content in *longissimus dorsi* muscles of Korean steers [13] and pigs [14]. Variation in *DGAT2* has been reported to affect carcass muscle and milk quality traits in various animals, such as fat yield and retail cut weight percentage in commercial feedlot steers [11], backfat thickness and lean percentage in pigs [15,16], milk yield and fat percentage in goats [17], and carcass weight, shear force, and IMF in domestic pigeons [18].

Yak (*Bos grunniens*) is indigenous to the area of Central Asia highlands and has been regarded as one of the world’s most remarkable domestic animal species, because it thrives in conditions of extreme harshness while providing meat, milk, fuel, and other necessities for local people. There are an estimated 15 million yaks in China, accounting for over 90% of the world’s yak population [19]. Yak meat, which is the staple animal protein food and the major source of income for the local herdsmen, is fine textured and high in protein and minerals [20]. Yak meat is regarded as being very palatable, but it is poor in IMF content and intramuscular marbling [21]. To date, little has been done to investigate the effect of the *DGAT2* gene on carcass muscle traits in yaks. The objective of this study was to screen genetic variation in yak *DGAT2* and to investigate whether the variation is associated with carcass and meat quality traits.

## 2. Materials and Methods

### 2.1. Animals and Sample Collection

All animal experiments were conducted in accordance with the guidelines for the care and use of experimental animals established by the Ministry of Science and Technology of the People’s Republic of China (Approval number 2006-398) and was approved by the Animal Care Committee of Gansu Agricultural University, Lanzhou, China.

A total of 694 Gannan yaks (one of the twelve officially recognized domestic yak breeds) were investigated. These yaks were randomly selected from the Gannan Tibetan Autonomous Prefecture, Gansu Province, China. All of these yaks grazed on the plateau pasture at an altitude of 2500–4000 m all year round. The gender and age of each yak was recorded before slaughter. At slaughter, a blood sample from each yak was collected onto a Flinders Technology Associates (FTA) card (Whatman BioScience, Middlesex, UK) to isolate genomic DNA using a two-step procedure described in Zhou et al. [22].

Twelve tissues, including heart, liver, spleen, lung, kidney, large intestine (colon), small intestine (jejunum), rumen, abomasum, mammary gland, *longissimus dorsi* muscle, and subcutaneous fat, were carefully collected from three five-year-old female Gannan yaks within 30 min of slaughter in October, which is considered to be the beginning of the cold season of the Qinghai-Tibetan Plateau. All samples were immediately snap-frozen in liquid nitrogen and then stored at −70 °C.

### 2.2. Carcass and Meat Quality Measurement

Carcass and meat quality traits were measured in the 694 Gannan yaks from which blood samples were collected. Hot carcass weight (HCW; kg) was measured just after slaughter. At 48 h postmortem, the rib eye area (REA; cm^2^) was traced on the cross-section of the *longissimus dorsi* muscle between twelfth and thirteenth rib of the right carcass side using sulfate paper, and the tracing area was determined by planimeter. The portion of longissimus muscle, taken from eleventh to thirteenth rib, was used to assess the meat quality. The Warner–Bratzler shear force (WBSF; kg), which represents meat tenderness, was estimated using a Digital Muscle Tenderometer (Model C-LM3, Northeast Agricultural University, Harbin, China), according to the methods of Shackelford et al. [23]. Cooking loss rate (CLR; %) and drip loss rate (DLR; %) were determined using the methods described by Honikel [24] and Liu et al. [25], respectively.

### 2.3. PCR-SSCP Analysis

Polymerase chain reaction (PCR) primers (see Table 1) were designed based on the bovine *DGAT2* sequence (GenBank Accession No. AC_000172.1) to amplify four fragments within intron 3–exon 4, intron 5, intron 6, and exon 8 regions in yak *DGAT2*. The primers were synthesized by Takara Biotechnology Co., Ltd. (Dalian, China).

PCR amplifications were run in a 20-μL reaction mixture, with the genome DNA on one 1.2-mm punch of blood spot on an FTA card as a template, 0.25 μM of each primer, 150 μM dNTP (Takara, Dalian, China), 2.5 mM Mg^2+^ and 0.5 U of *Tαq* DNA polymerase (Takara, Dalian, China), and double-distilled water to a final volume of 20 μL. The cycling protocols were 2 min at 94 °C, followed by 35 cycles of 30 s at 94 °C, 30 s at the annealing temperatures (Table 1), and 30 s at 72 °C, and a final extension of 5 min at 72 °C.

A single-stranded conformational polymorphism (SSCP) method was used to screen variation in each of the regions amplified. An aliquot of 2-μL amplicon was mixed with 8 μL of loading dye (98% formamide, 10 mM ethylenediaminetetraacetic acid (EDTA), 0.025% bromophenol blue, 0.025% xylene cyanol). The mixtures were denatured at 98 °C for 5 min, then chilled rapidly on wet ice for 5 min and subjected to 14% polyacrylamide (39:1) gels. Samples were electrophoresed in 0.5 × Tris-boric-acid-EDTA (TBE) buffer for 18 h under conditions described in Table 1. The gels were silver-stained according to Byun et al. [26].

### 2.4. Variant Sequencing and Analysis

Amplicons that were confirmed as homozygous by SSCP were directly sequenced in both directions at Sangon Biotech (Shanghai, China). Variants that were only found in heterozygous yaks were sequenced using a rapid sequencing method described by Hu et al. [27]. Sequence alignments were performed by DNAMAN (version 5.2.10, Lynnon BioSoft, Vaudreuil, QC, Canada). 

### 2.5. Haplotype Determination

Haplotypes were constructed across introns 5 and 6 of *DGAT2*, as variations were only found in these two regions. For yaks that were identified as homozygous in either region, it was possible to directly ascertain their haplotypes based on the co-inheritance of sequences. For example, if a yak presented with genotype *A_1_A_1_* in intron 5 and genotype *A_2_B_2_* in intron 6, the presence of haplotypes *A_1_–A_2_* and *A_1_–B_2_* could be directly inferred. For those yaks that were heterozygous in both regions, the haplotypes could not be inferred because their parents’ genotypes of *DGAT2* were not available for analysis in this study.

### 2.6. Statistical Analyses

Association analyses of *DGAT2* variation in relation to carcass and meat quality traits of yak were performed with the general linear mixed models (GLMMs) in IBM SPSS 24.0 (IBM Corp., Armonk, NY, USA). Initially, single-variant/haplotype models were conducted to confirm which variants/haplotypes should be factored into subsequent multi-variant models that would include any variant/haplotype that had an association with a trait in the single-variant/haplotype models with *p* < 0.2, and which might therefore potentially impact on the carcass and meat quality traits being tested.

Gender, age, and group were found to affect carcass and meat quality traits, and they were therefore fitted into the models. Unless otherwise indicated, all *p*-values less than 0.05 were considered to be significantly different.

### 2.7. RNA Extraction and RT-qPCR Analysis

The reverse transcription-qPCR (RT-qPCR) was performed to ascertain the yak *DGAT2* expression in tissues using yak *β-actin* as an internal control. Two pairs of qPCR primers were designed to amplify a 149-bp fragment of yak *DGAT2* and a 133-bp fragment of *β-actin* based on the mRNA sequences (GenBank Accession Nos. XM_005902498 and DQ838049.1, respectively). The *DGAT2* qPCR primers were 5’-GGCCTCTTCTCCTCTGACACC-3’ and 5’-CACCAGGGCTTGCACGTACA-3’, and the β-actin primers were 5’-AGCCTTCCTTCCTGGGCATGGA -3’ and 5’- GGACAGCACCGTGTTGGCGTAGA -3’. Total RNA was extracted from tissue samples of yak using TRIzol reagent (Invitrogen, Carlsbad, CA, USA) according to the manufacturer’s instructions. The integrity and concentration of total RNA was assessed by 2% agarose gels in electrophoresis and UV spectrophotometry. The cDNA was synthesized by reverse transcription from total RNA using the PrimeScript^TM^ RT Reagent Kit with the gDNA Eraser (Takara) following the manufacturer’s instructions. The amplification of the cDNA was conducted in 20-μL reaction mixture consisting of 100 ng cDNA, 0.25 μM of each primer, 10.0 μL AceQ qPCR SYBR^®^ Green Master Mix (Vazyme, Nanjing, China), 0.4 μL ROX Reference Dye 2, and supplemented with ddH_2_O to a volume of 20 μL. The thermal profiles were one cycle of 5 min at 95 °C, followed by 40 PCR cycles of 10 s at 95 °C, 30 s at 60 °C, and 30 s at 72 °C. The 2^−ΔΔ*C_T_*^ method was used to analyze the relative expression data [28]. Amplification was performed in Applied Biosystems Quant Studio^®^6 Flexq (Applied Biosystems, Carlsbad, CA, USA).

## 3. Results

### 3.1. Identification of Variation in Yak DGAT2

The four pairs of PCR primers, which amplified four fragments of yak *DGAT2*, including the intron 3*–*exon 4, intron 5, intron 6, and exon 8 fragments, worked well on all of the DNA samples under the conditions established. In the 694 yaks investigated, three PCR-SSCP patterns representing three variant sequences were detected for both introns 5 and 6, respectively (Figure 1A). Four single-nucleotide polymorphisms (SNPs) were detected among those sequences (Figure 1B). Of these, SNPs c.612 + 350C/A and c.612 + 228A/G were located in intron 5 and c.787 + 616G/A and c.787 + 691T/G in intron 6. No variations were detected for intron 3–exon 4 and exon 8 fragments (Figure 1A).

### 3.2. Polymorphisms in Yak DGAT2

In the 694 yaks, all genotype and variant frequencies in each region were >5% (Table 2). Variants *A_1_* (in intron 5) and *A_2_* (in intron 6) were the most common, with frequencies of 69.96% and 65.49%, respectively.

Seven haplotypes, spanning introns 5 and 6 of *DGAT2*, could be determined in the 469 individuals that were a subset of the 694 yaks in which the variants were identified. Of these haplotypes, *A_1_*-*A_2_*, *A_1_*-*B_2_*, and *B_1_*-*A_2_* were the most common, with frequencies of 64.71%, 11.62%, and 14.18%, respectively. The other four haplotypes, *A_1_*-*C_2_*, *B_1_*-*B_2_*, *C_1_*-*A_2_*, and *C_1_*-*B_2_* were rare, with frequencies of 3.30%, 0.64%, 4.37%, and 1.17%, respectively.

### 3.3. Associations of DGAT2 Variation with Carcass and Meat Quality Traits in Yak

In the single-variant (presence/absence) model (Table 3), the presence of variant *B_1_* and the absence of *C_1_* in intron 5 were associated with a decrease in WBSF (*p* = 0.0020 and *p* = 0.0434, respectively), and the presence of *C_1_* was associated with decreased DLR (*p* = 0.0271). The presence of *C_2_* and *B_2_* in intron 6 were found to be associated with a decrease in WBSF (*p* = 0.0441) and CLR (*p* = 0.0142), respectively. These associations with WBSF remained significant when the other variants were factored into the models (*p* < 0.2). No associations with REA and HCW were detected for any variant in both regions of yak *DGAT2*.

In the single-haplotype (presence/absence) model (Table 4), the absence of haplotype *A_1_*-*A_2_* and the presence of haplotype *B_1_*-*A_2_* were associated with a decrease in WBSF (*p* = 0.0191 and *p* = 0.0010, respectively). These associations persisted in the multi-variant models. No effects of *DGAT2* haplotypes were found on HCW, REA, DLR and CLR.

### 3.4. Tissue Expression of Yak DGAT2

The *DGAT2* expression level was significantly different among twelve tissues of yak (*p* < 0.05; Figure 2). The highest expression was found in subcutaneous fat, whereas moderate expression was detected in the liver, heart, *longissimus dorsi* muscle, and abomasum, and weak expression was observed in other tissues, including the large intestine, small intestine, mammary gland, lungs, kidney, spleen, and rumen.

## 4. Discussion

This is the first report to elucidate the associations between *DGAT2* variation and meat tenderness in yak. Meat tenderness, which is considered to be the most important component of beef quality identified by consumers, is positively influenced by IMF content [29,30]. IMF could improve meat tenderness by weakening structures of intramuscular connective tissue [31]. Hocquette et al. [29] found that the IMF content had a positive genetic correlation with tenderness (r_G_ = 0.41) and a negative correlation with WBSF (r_G_ = −0.50) in cattle. Given that the deposition of IMF is mainly determined by lipid metabolism, many lipid metabolic genes may be involved in fat deposition in muscle [13]. *DGAT2* plays an important role in fat deposition, as it is a key enzyme that catalyzes the final and rate-limited step of TG synthesis. Wakimoto et al. [32] found that *DGAT2* is associated with adipose assembly in adipose tissue and skeletal muscle, and Winter et al. [10] reported that *DGAT2* could be a priority candidate gene for quantitative traits related to TG synthesis and storage in farm animals.

In this study, a total of four SNPs were detected in introns 5 and 6 of the yak *DGAT2* gene. The level of variation detected in yaks appears to be lower than that reported in cattle in which 13 SNPs and 2 ins/del variations have been found in introns 5 and 6 [10]. SNPs in introns have been shown to affect gene expression by regulating the rate of transcription, nuclear export, transcript stability, and the efficiency of mRNA translation in many eukaryotes [33]. Moreover, the optimal expression of many genes also requires the presence of one or more introns [34]. Although these SNPs in yak *DGAT2* were not located in the coding regions, they may influence the gene expression, and thus affect fat-related traits.

The association between variation in *DGAT2* and the various quantitative traits related to TG synthesis and storage has been recently reported in a number of other animal species. Li et al. [11] found that SNPs in intron 3 of *DGAT2* were associated with the body weight and fat yield in commercial feedlot steers. Yin et al. [15] and Li et al. [16] reported that a SNP and 13 bp indel in the 3′-UTR of *DGAT2* were associated with backfat thickness and lean percentage in pigs. Mao et al. [18] found that SNPs in exons 5 and 6 of *DGAT2* had an effect on carcass and meat quality traits in domestic pigeons. In goats, variations in exon 3 of *DGAT2* were found to be associated with growth traits [35], while variations in exon 4 and intron 5 had an influence on the milk yield and fat percentage [17]. In current study, variations in introns 5 and 6 of yak *DGAT2* were identified and showed that the presence of variant *B_1_* and absence of *C_1_* in intron 5, as well as the presence of *C_2_* in intron 6, was associated with a decrease in WBSF. Despite there being few reports on the associations between *DGAT2* variation and WBSF in other species, the level of *DGAT2* expression was found to be positively correlated (*p* < 0.05) with IMF content in cattle and pigs [13,14]. *DGAT2* was found to be moderately expressed in *longissimus dorsi* muscle of yak in this study. Given the positive influence of the IMF content on meat tenderness, variation in yak *DGAT2* could affect meat tenderness by regulating IMF deposition in muscle.

In this study, the absence of haplotype *A_1_*-*A_2_* and the presence of *B_1_*-*A_2_* were associated with a decrease in WBSF of yak meat. Moreover, no haplotype was found to be associated with DLR and CLR, even though the presence of variant *C_1_* in intron 5 and *B_2_* in intron 6 was associated with decreased DLR and increased CLR, respectively. Haplotypes are particular combinations of alleles or variant sequences on a single chromosome. Their information, which is composed of multiple markers, would be more powerful than individual and unorganized markers, as they may improve the power of genome-wide association studies [36,37]. In general, molecular markers could be accurately identified when integrating the haplotypes and SNPs into association analysis [38]. For instance, Zollner et al. [39] reported that markers with more variants were much more efficient in detecting any significant associations. The associations identified here suggest that variations in yak *DGAT2* affect meat tenderness. Further investigations on more yaks from different farms and breeds are needed to confirm this. 

*DGAT2* is expressed in a variety of tissues in mammals. In this study, we found that the highest expression level of yak *DGAT2* was in subcutaneous fat, followed by the liver, heart, *longissimus dorsi* muscle, and other tissues. High expression of *DGAT2* in adipose tissue indicates an important function in triglyceride storage [4]. Yaks are indigenous to the Qinghai-Tibet Plateau and adjacent highlands at an altitude from 2000 m to 5000 m with cold, semi-humid climates, and they deposit more subcutaneous fat prior to winter in order to cope with the severe cold [21]. In addition, previous studies have showed that the TG synthesis pathway is not an important step for hepatic fat accumulation in cattle [40,41]. Therefore, we speculate that the higher *DGAT2* expression in subcutaneous fat than in liver is propitious to more accumulation of subcutaneous fat, which would provide more energy reserve that can used over the seven-month long cold season in the Qinghai-Tibetan Plateau. However, some reports have showed that *DGAT2* expression abundance might relate to the species and developmental stage of tissues. The highest expression of *DGAT2* is found in adipose tissues in mice, but in pigs and humans the highest level of expression is in the liver [4,14]. In sheep, different expression levels are observed in various tissues at different developmental stages, such as the liver, rumen, and jejunum [42].This discrepancy may be due to the biological function of *DGAT2*, and it is speculated that the post-transcriptional regulatory mechanism plays a role in *DGAT2* expression in different species and developmental stages. Further studies are required to detect *DGAT2* expression in tissues from yaks at different development stages and with different *DGAT2* variants in order to understand whether genetic variations and developmental stages affect gene expression.

## 5. Conclusions

This study elucidated tissue expression of the yak *DGAT2* gene and revealed variation in the gene. The highest expression was found in the subcutaneous fat of yaks. Variants and haplotypes in introns 5 and 6 of yak *DGAT2* were associated with the WBSF of longissimus muscle. These results suggest that *DGAT2* variations could be used as genetic markers in breeding programs to improve meat tenderness in yaks.

## Figures and Tables

**Figure 1 animals-09-00061-f001:**
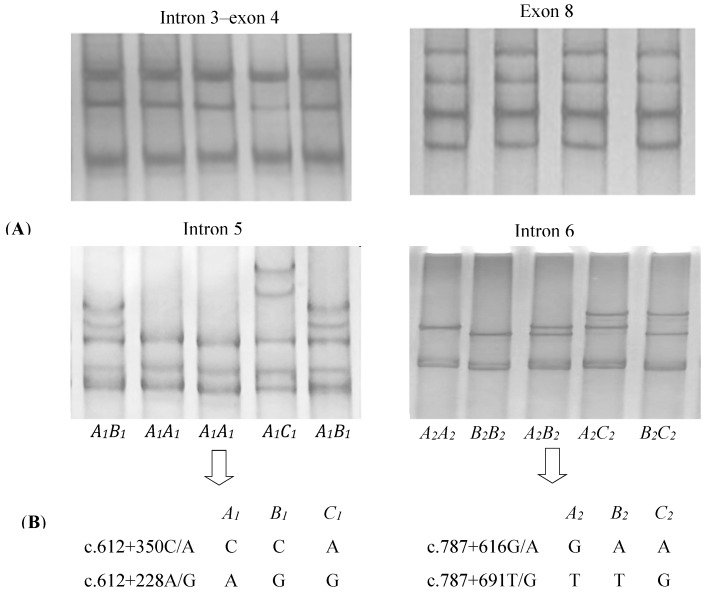
SSCP (single-stranded conformational polymorphism) banding patterns for the four regions of yak *DGAT2* amplified (**A**) and single-nucleotide polymorphisms (SNPs) detected in introns 5 and 6 of yak *DGAT2* (**B**).

**Figure 2 animals-09-00061-f002:**
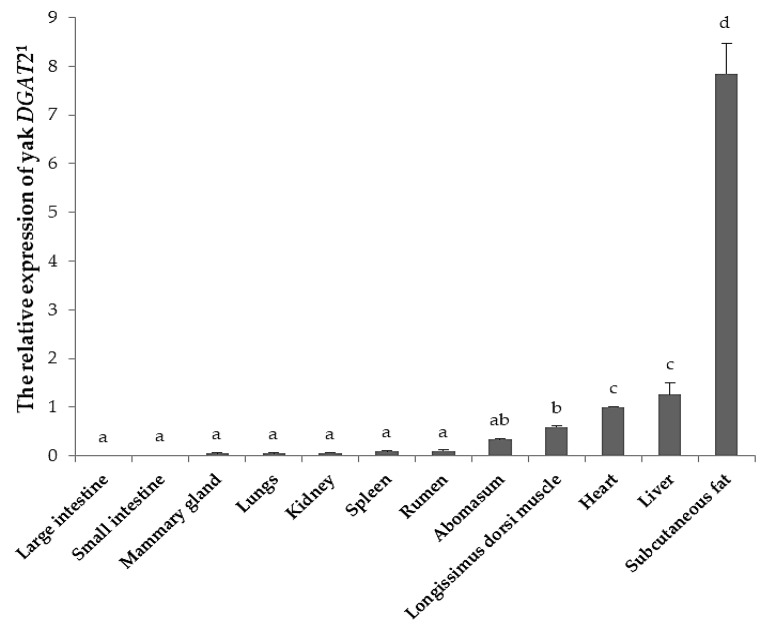
The relative expression of *DGAT2* in twelve tissues of yak. ^1^ Heart represents control tissue. Error bars indicates standard deviation. The different lower-case letters on columns indicate significant difference (*p* < 0.05).

**Table 1 animals-09-00061-t001:** PCR primers sequences and SSCP conditions for yak *DGAT2*.

Region	Primer Sequence (5′→3′)	Amplicon Size (bp)	Annealing Temperature	SSCP Condition
Intron 3−exon 4	TAAGCCTGGGCATGGTTC	344	61 °C	220 V, 14%, 15 °C
	CCTCCCAAGATAACACCTGC			
Intron 5	TAGGAAACCTTCTCTGACCC	288	64 °C	260 V, 14%, 12 °C
	CAGCCACTTAGAAGAACAGC			
Intron 6	CCTATGCCAAAGCCTGTCAC	459	64 °C	240 V, 14%, 10 °C
	CCCAGACACCAGCCAAACT			
Exon 8	CACATCTGGGCCTTTATG	229	63 °C	220 V, 14%, 15 °C
	CTTGGCAAGAGGGTTTAGTC			

SSCP: single-stranded conformational polymorphism.

**Table 2 animals-09-00061-t002:** Genotype and variant frequency of yak *DGAT2*.

Genotype Frequency (%)								Variant Frequency (%)					
Intron 5			Intron 6					Intron 5			Intron 6		
*A_1_A_1_*	*A_1_B_1_*	*A_1_C_1_*	*A_2_A_2_*	*B_2_B_2_*	*A_2_B_2_*	*A_2_C_2_*	*B_2_C_2_*	*A_1_*	*B_1_*	*C_1_*	*A_2_*	*B_2_*	*C_2_*
39.91	42.24	15.85	47.41	24.35	7.35	11.82	9.08	69.96	22.12	7.92	65.49	24.06	10.45

**Table 3 animals-09-00061-t003:** Association of *DGAT2* variants with carcass and meat quality traits (mean ± *SE*) ^1^ in yaks.

Region	Trait (unit) ^2^	Variant	Other Variants in Model	*n*		Mean ± *SE*		*p*-Value
				Present	Absent	Present	Absent	
Intron 5	HCW (kg)	*B_1_*		417	277	105.16 ± 2.89	107.53 ± 3.01	0.4996
		*C_1_*		110	584	101.52 ± 4.55	106.89 ± 2.42	0.2223
	REA (cm^2^)	*B_1_*		417	277	32.06 ± 0.54	32.00 ± 0.59	0.9201
		*C_1_*		110	584	31.20 ± 0.86	32.16 ± 0.49	0.2431
	WBSF (kg)	*B_1_*		417	277	5.12 ± 0.09	5.44 ± 0.10	0.0020
		*C_1_*		110	584	5.50 ± 0.14	5.22 ± 0.08	0.0434
		*B_1_*	*C_1_*	417	277	5.25 ± 0.10	5.70 ± 0.12	0.0001
		*C_1_*	*B_1_*	110	584	5.72 ± 0.15	5.23 ± 0.08	0.0008
	DLR (%)	*B_1_*		417	277	21.75 ± 0.38	22.23 ± 0.41	0.2643
		*C_1_*		110	584	20.86 ± 0.59	22.12 ± 0.34	0.0271
	CLR (%)	*B_1_*		417	277	33.89 ± 0.39	34.15 ± 0.42	0.5593
		*C_1_*		110	584	33.96 ± 0.62	34.01 ± 0.35	0.9294
Intron 6	HCW (kg)	*A_2_*		604	90	106.74 ± 2.42	102.10 ± 4.95	0.3383
		*B_2_*		259	435	103.92 ± 3.18	107.66 ± 2.68	0.2711
		*C_2_*		145	549	106.39 ± 3.83	106.24 ± 2.50	0.9695
	REA (cm^2^)	*A_2_*		604	90	32.06 ± 0.48	31.81 ± 0.95	0.7836
		*B_2_*		259	435	31.76 ± 0.62	32.19 ± 0.53	0.4949
		*C_2_*		145	549	32.90 ± 0.79	31.87 ± 0.49	0.1691
	WBSF (kg)	*A_2_*		604	90	5.27 ± 0.08	5.20 ± 0.16	0.6429
		*B_2_*		259	435	5.37 ± 0.10	5.20 ± 0.09	0.1132
		*C_2_*		145	549	5.05 ± 0.13	5.30 ± 0.08	0.0441
		*B_2_*	*C_2_*	259	435	5.29 ± 0.11	5.10 ± 0.10	0.0853
		*C_2_*	*B_2_*	145	549	5.06 ± 0.13	5.33 ± 0.08	0.0339
	DLR (%)	*A_2_*		604	90	21.96 ± 0.33	21.94 ± 0.66	0.9752
		*B_2_*		259	435	21.54 ± 0.43	22.19 ± 0.36	0.1327
		*C_2_*		145	549	22.61 ± 0.54	21.83 ± 0.34	0.1288
	CLR (%)	*A_2_*		604	90	34.04 ± 0.34	33.60 ± 0.68	0.4966
		*B_2_*		259	435	33.30 ± 0.44	34.40 ± 0.37	0.0142
		*C_2_*		145	549	34.50 ± 0.56	33.90 ± 0.35	0.2606

^1^ Estimated means, their standard errors (*SE*s), and *p*-values derived from general linear mixed models. ^2^ HCW represents hot carcass weight, REA represents rib eye area, WBSF represents Warner–Bratzler shear force, DLR represents drip loss rate, and CLR represents cooking loss rate.

**Table 4 animals-09-00061-t004:** Association of *DGAT2* haplotypes with carcass and meat quality traits (mean ± *SE*) ^1^ in yaks.

Trait (unit) ^2^	Haplotype	Other Haplotypes in Model	*n*		Mean ± *SE*		*p*-Value
			Present	Absent	Present	Absent	
HCW (kg)	*A_1_*-*A_2_*		428	41	107.90 ± 2.82	94.74 ± 8.63	0.1234
	*A_1_*-*B_2_*		99	370	99.74 ± 5.20	109.33 ± 3.01	0.0832
	*A_1_*-*C_2_*		31	438	108.42 ± 11.95	107.35 ± 2.83	0.9281
	*B_1_*-*A_2_*		133	336	108.81 ± 4.72	106.90 ± 3.08	0.7051
	*C_1_*-*A_2_*		41	428	100.04 ± 11.86	107.63 ± 2.85	0.5254
REA (cm^2^)	*A_1_*-*A_2_*		428	41	31.89 ± 0.58	33.11 ± 1.40	0.3644
	*A_1_*-*B_2_*		99	370	32.38 ± 0.93	31.83 ± 0.61	0.5587
	*A_1_*-*C_2_*		31	438	34.19 ± 1.58	31.87 ± 0.57	0.1283
	*B_1_*-*A_2_*		133	336	32.05 ± 0.83	31.91 ± 0.62	0.8651
	*C_1_*-*A_2_*		41	428	32.97 ± 1.35	31.84 ± 0.59	0.4051
WBSF (kg)	*A_1_*-*A_2_*		428	41	5.29 ± 0.10	4.76 ± 0.23	0.0191
	*A_1_*-*B_2_*		99	370	5.24 ± 0.16	5.27 ± 0.10	0.8518
	*A_1_*-*C_2_*		31	438	5.28 ± 0.27	5.26 ± 0.10	0.9506
	*B_1_*-*A_2_*		133	336	4.93 ± 0.14	5.39 ± 0.10	0.0010
	*C_1_*-*A_2_*		41	428	5.48 ± 0.23	5.24 ± 0.10	0.2903
	*A_1_*-*A_2_*	*B_1_*-*A_2_*	428	41	5.18 ± 0.10	4.48 ± 0.24	0.0021
	*B_1_*-*A_2_*	*A_1_*-*A_2_*	133	336	4.56 ± 0.18	5.10 ± 0.14	0.0001
DLR (%)	*A_1_*-*A_2_*		428	41	22.23 ± 0.41	22.20 ± 0.99	0.9756
	*A_1_*-*B_2_*		99	370	22.37 ± 0.66	22.19 ± 0.43	0.7854
	*A_1_*-*C_2_*		31	438	23.51 ± 1.12	22.18 ± 0.41	0.2186
	*B_1_*-*A_2_*		133	336	22.71 ± 0.59	22.04 ± 0.44	0.2580
	*C_1_*-*A_2_*		41	428	20.83 ± 0.96	22.38 ± 0.42	0.1055
CLR (%)	*A_1_*-*A_2_*		428	41	34.02 ± 0.40	34.03 ± 0.98	0.9925
	*A_1_*-*B_2_*		99	370	33.37 ± 0.65	34.20 ± 0.42	0.2058
	*A_1_*-*C_2_*		31	438	34.66 ± 1.11	34.00 ± 0.40	0.5360
	*B_1_*-*A_2_*		133	336	34.17 ± 0.58	33.97 ± 0.43	0.7337
	*C_1_*-*A_2_*		41	428	33.51 ± 0.94	34.08 ± 0.41	0.5513

^1^ Estimated means, their standard errors (*SE*s), and *p*-values derived from general linear mixed models. ^2^ HCW represents hot carcass weight, REA represents rib eye area, WBSF represents Warner–Bratzler shear force, DLR represents drip loss rate, and CLR represents cooking loss rate.

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
