# Peer review of "Tissue Expression and Variation of the DGAT2 Gene and Its Effect on Carcass and Meat Quality Traits in Yak"

_animals, 2019, doi:10.3390/ani9020061_

Round 1

Reviewer 1 Report

In this manuscript the expression of DGAT2 in a set of twelve different tissues from yak is reported. DGAT2 has been shown to regulate fat deposition in livestock, e.g. cattle and pigs, and as such the authors also test variation in DGAT2 on carcass and meat quality, by measuring hot carcass weight, cooking loss rate, drip loss weight and Warner-Brazler shear force on longissimus dorsi muscle.

The manuscript is well written and the conclusions clear but it could do with some re-structuring and clarification on a few points in the methodology.

Methods

If I understand correctly the DNA samples were collected from a random subset of 694 yaks being processed through slaughterhouse while the tissue samples were from Gannan yaks? How many ‘breeds’ of Yak are likely to be included in the sample of 694 yaks and how prevalent were the haplotypes detected across ‘breeds’? Or were all 694 individuals Gannan yaks?

The section on statistical analyses could be moved to come before the section on qPCR so it’s clear that this analysis is performed on the 694 animals from the slaughterhouse and not the 3 animals from which the tissues were collected.

Results

The results section doesn’t follow the same order as the methods section, it would be clearer for the reader if the order was the same in both, i.e. move expression of DGAT2 across tissues to the end of the results section rather than beginning with it.

Given DGAT2 expression in low in muscle, certainly relative to fat, this indicates that DGAT2 expression might have more an effect on fat than muscle traits, were any fat traits measured?

It’s unclear whether the individuals used to test for variation and to assess haplotype prevalence are the same animals and the number of animals that were used to assess the carcass quality and other traits. Although this does become clearer on reaching tables 3-4. Could the authors please clarify these points in the methods so it is clear from the beginning?

Discussion

Similarly the order of the discussion doesn’t reflect the methods section and it might be clearer to move the discussion of tissue specific expression to the end.

The authors speculate that there might be interesting species specific expression of DGAT2 related to fat accumulation and adaptation to altitude and cold climate. Considerably more work is required to test this hypothesis but how this might be achieved and what novel applications might be could be added to the discussion.

Presumably an important next step would be to measure DGAT2 expression in tissues e.g. liver, muscle and fat from more individuals with DGAT2 variants to test whether these variants affect gene expression but this is not mentioned in the discussion. 

Line Changes

Line 49 Remove ‘quantity’

Line 62-62 Change ‘variations’ to ‘variants’

Line 72 Change ‘were’ to ‘was’

Line 73 Change ‘was’ to ‘were’

Line 75 Which region of the small and large intestine were sampled?

Line 111 Change ‘by’ to ‘using’ and please clarify the method of sequencing here as well as the reference.

Line 113 Change ‘haplotypes’ to ‘haplotype’

Line 122 Insert ‘an’ before internal control

Line 126 Change ‘were’ to ‘was’

Figure 1 could be improved by labeling the tissues rather than providing a list, numbering the list would also help to make the figure clearer.

Line 155 Remove ‘were’

Line 156 Insert ‘the’ before ‘conditions’

Line 181 469 or 694?? Presumably 469 from tables 3-4 and haplotype prevalence was assessed in a smaller subset of animals. Are these individuals a subset of the 694 in which the variants are assessed? This is unclear from the methods section.

Line 191 Change ‘significantly’ to ‘significant’

Line 221 The discrepancy might also relate to the developmental stage sampled. DGAT2 might be higher in the liver or younger yaks for example. How do the expression levels of DGAT2 in yak liver correspond to other ruminants? E.g. cattle or sheep – the authors could check the expression of DGAT2 across tissues in sheep using the sheep gene expression atlas on BioGPS http://www.biogps.org/sheepatlas

Line 236-237 I think it is the same thing if the variant has a regulatory role affecting gene expression or if it directly affects gene expression?

Line 251-253 As IMF hasn’t been measured in this study it might be better to adjust this sentence to ‘could affect’ more than ‘most likely affects’.

Line 259-263 These sentences are unclear, could they be reworded for clarity? Given the evidence presented only indicates that the variants might affect meat tenderness in yak it would be unwise to suggest breeding away from these haplotypes without further testing. Perhaps rewording to indicate that after further testing this might be a suitable strategy would solve this? 

Author Response

Response to Reviewer 1 Comments

Methods

If I understand correctly the DNA samples were collected from a random subset of 694 yaks being processed through slaughterhouse while the tissue samples were from Gannan yaks? How many ‘breeds’ of Yak are likely to be included in the sample of 694 yaks and how prevalent were the haplotypes detected across ‘breeds’? Or were all 694 individuals Gannan yaks?

Gannan yak is one of 12 officially recognized breeds of domestic yak that distributed in the Gannan Tibetan Autonomous Prefecture, Gansu Province, China. In this study, all of the DNA samples were collected from the Gannan yak. Seven haplotypes were determined in the 469 individuals that were a subset of the 694 Gannan yaks in which the variants were identified. We have clarified these in the method and result sections in revised manuscript.

The section on statistical analyses could be moved to come before the section on qPCR so it’s clear that this analysis is performed on the 694 animals from the slaughterhouse and not the 3 animals from which the tissues were collected.

The section of statistical analyses has been moved to come before the section on qPCR, as suggested.

Results

The results section doesn’t follow the same order as the methods section, it would be clearer for the reader if the order was the same in both, i.e. move expression of DGAT2 across tissues to the end of the results section rather than beginning with it.

We have moved expression of DGAT2 across tissues to the end of the results section in revised manuscript.

Given DGAT2 expression in low in muscle, certainly relative to fat, this indicates that DGAT2 expression might have more an effect on fat than muscle traits, were any fat traits measured?

A very good suggestion, but unfortunately no any fat traits of these Gannan yaks, such as subcutaneous fat, intramuscular fat content and fatty acid composition, has been measured.

It’s unclear whether the individuals used to test for variation and to assess haplotype prevalence are the same animals and the number of animals that were used to assess the carcass quality and other traits. Although this does become clearer on reaching tables 3-4. Could the authors please clarify these points in the methods so it is clear from the beginning?

The yaks used for assessing the carcass quality and other traits were the same as those that were used to test for variation and to infer haplotypes, but the haplotypes were only determined in a subset (469 yaks) of these samples (694 yaks). We have clarified these in the method section in revised manuscript.

Discussion

Similarly the order of the discussion doesn’t reflect the methods section and it might be clearer to move the discussion of tissue specific expression to the end.

The discussion of tissue specific expression has been moved to the end in revised manuscript.

The authors speculate that there might be interesting species specific expression of DGAT2 related to fat accumulation and adaptation to altitude and cold climate. Considerably more work is required to test this hypothesis but how this might be achieved and what novel applications might be could be added to the discussion.

Hypothesis that DGAT2 expression related to cold adaptation via fat accumulation depend on gene function in triglyceride storage and characteristic of yak that could deposit more subcutaneous fat prior to winter. In view of complexity of yak adaptions for cold climate, more work is required to confirm it, but this is probably outside the scope of this study.

Presumably an important next step would be to measure DGAT2 expression in tissues e.g. liver, muscle and fat from more individuals with DGAT2 variants to test whether these variants affect gene expression but this is not mentioned in the discussion. 

We mentioned this point at the end of discussion section in revised manuscript.

Line Changes

Line 49 Remove ‘quantity’

Corrected.

Line 62-62 Change ‘variations’ to ‘variants’

We have reworded this sentence in revised manuscript.

Line 72 Change ‘were’ to ‘was’

Corrected.

Line 73 Change ‘was’ to ‘were’

Corrected.

Line 75 Which region of the small and large intestine were sampled?

Corrected.

Line 111 Change ‘by’ to ‘using’ and please clarify the method of sequencing here as well as the reference.

Corrected.

Line 113 Change ‘haplotypes’ to ‘haplotype’

Corrected.

Line 122 Insert ‘an’ before internal control

Corrected.

Line 126 Change ‘were’ to ‘was’

Corrected.

Figure 1 could be improved by labeling the tissues rather than providing a list, numbering the list would also help to make the figure clearer.

Corrected and changed “Figure1” to “Figure 2”.

Line 155 Remove ‘were’

Corrected.

Line 156 Insert ‘the’ before ‘conditions’

Corrected.

Line 181 469 or 694?? Presumably 469 from tables 3-4 and haplotype prevalence was assessed in a smaller subset of animals. Are these individuals a subset of the 694 in which the variants are assessed? This is unclear from the methods section.

Haplotype prevalence was assessed in 469 individuals that are a subset of 694 yaks in which the variant were detected. We have clarified this in3.2 Polymorphisms in yak DGAT2.

Line 191 Change ‘significantly’ to ‘significant’

Corrected.

Line 221 The discrepancy might also relate to the developmental stage sampled. DGAT2 might be higher in the liver or younger yaks for example. How do the expression levels of DGAT2 in yak liver correspond to other ruminants? E.g. cattle or sheep – the authors could check the expression of DGAT2 across tissues in sheep using the sheep gene expression atlas on BioGPS http://www.biogps.org/sheepatlas  

A very good suggestion, we have further discussed expression of DGAT2 in species specific and developmental stage of tissue in revised manuscript.

Line 236-237 I think it is the same thing if the variant has a regulatory role affecting gene expression or if it directly affects gene expression?

We have reworded this sentence in revised manuscript.

Line 251-253 As IMF hasn’t been measured in this study it might be better to adjust this sentence to ‘could affect’ more than ‘most likely affects’.

Corrected.

Line 259-263 These sentences are unclear, could they be reworded for clarity? Given the evidence presented only indicates that the variants might affect meat tenderness in yak it would be unwise to suggest breeding away from these haplotypes without further testing. Perhaps rewording to indicate that after further testing this might be a suitable strategy would solve this? 

We have reworded these sentences in revised manuscript.

Reviewer 2 Report

This is a well written manuscript and well conducted study regarding the identification of putative biomarkers for the improvement of meat quality traits in yak. The tables and figures are professionally presented. There are some comments that have to be addressed, though, before the manuscript is accepted for publication.

1. The introduction is too short. The authors should provide literature papers with respect to the significance of yak as a source of nutrients for the people of central Asia. Are there any studies presenting specific biological traits of yak or the products generated from it (i.e., protective against diseases, antioxidant etc)?

2. The authors state that: "Our results suggested that DGAT2 variation could be used as genetic markers in breeding programs to improve meat tenderness in yak." How will this happen in practice? How reliable biomarkers of meat quality are the DGAT2 variations? Could they be used as biomarkers for products generated from yak (i.e., milk), too?   

3. Do the authors believe that there are specific genetic variations of DGAT2 that are superior compared to others presented in the manuscript?

4. Figure 1. How did the authors calculate the relative expression of yak DGAT2?

Author Response

Response to Reviewer 2 Comments

         1. The introduction is too short. The authors should provide literature papers with respect to the significance of yak as a source of nutrients for the people of central Asia. Are there any studies presenting specific biological traits of yak or the products generated from it (i.e., protective against diseases, antioxidant etc)?

        Literature papers have been provided to clarify further the yak population size, meat quality, and essentiality for local people in the introduction part in revised manuscript.

        2. The authors state that: "Our results suggested that DGAT2 variation could be used as genetic markers in breeding programs to improve meat tenderness in yak." How will this happen in practice? How reliable biomarkers of meat quality are the DGAT2 variations? Could they be used as biomarkers for products generated from yak (i.e., milk), too?   

        In this study, we assessed the effect of DGAT2 on meat quality traits depending on relative large sample size of Gannan yak, which is one of 12 officially recognized breeds of domestic yaks. Those results provide a potential method of molecular assisted selection that can predict the meat tenderness in Gannan yak breeding practice. Certainly, further studies are needed to confirm those results in larger sample size of yaks, including other yak breeds.

        In many cases, DGAT2 is a candidate gene that affects meat and milk quality traits of livestock. There are, however, few reports about associations between DGAT2 and products, meat and milk quality traits of yak. So, more researches would be required to test whether these variants, which affected meat tenderness of Gannan yak in this study, could be used as genetic markers for products or other fat-related traits of yak (i.e., milk quality traits) in future.

        3. Do the authors believe that there are specific genetic variations of DGAT2 that are superior compared to others presented in the manuscript?

DGAT2 is associated with various fat related-traits in animals. According to the association results obtained in this study, meat of Gannan yaks with haplotype B1-A2 or without haplotype A1-A2 of DGAT2 was more tender than meat of Gannan yaks with other haplotypes. Other genetic variation effects on other fat related-traits of yak need to be investigated furtherly.

4. Figure 1. How did the authors calculate the relative expression of yak DGAT2?

The 2-ΔΔ CT method (Livak et al., 2001” in References) was used to calculate the relative expression of yak DGAT2.

Round 2

Reviewer 2 Report

The authors have successfully addressed my comments.